# Effect of Steel Fibre Reinforcement on Flexural Fatigue Behaviour of Notched Structural Concrete

**DOI:** 10.3390/ma14195854

**Published:** 2021-10-06

**Authors:** Jose A. Sainz-Aja, Laura Gonzalez, Carlos Thomas, Jokin Rico, Juan A. Polanco, Isidro Carrascal, Jesús Setién

**Affiliations:** 1LADICIM (Laboratory of Materials Science and Engineering), E.T.S. de Ingenieros de Caminos, Canales y Puertos, University of Cantabria, Av./Los Castros 44, 39005 Santander, Spain; jose.sainz-aja@unican.es (J.A.S.-A.); juan.polanco@unican.es (J.A.P.); isidro.carrascal@unican.es (I.C.); jesus.setien@unican.es (J.S.); 2INGECID S.L. (Ingeniería de la Construcción, Investigación y Desarrollo de Proyectos), E.T.S. de Ingenieros de Caminos, Canales y Puertos, University of Cantabria, Av./Los Castros 44, 39005 Santander, Spain; laura.gonzalez@ingecid.es (L.G.); jokinrico@ingecid.es (J.R.)

**Keywords:** bending fatigue, fibre-reinforced concrete, S-N curve, Wöhler curve, concrete damage

## Abstract

One of the biggest challenges in facilitating the installation of concrete is the development of fibre-reinforced concrete. Although nowadays fibre reinforced concrete is relatively common, it is still necessary to deepen in the study on its behaviour, especially regarding its fatigue behaviour. This paper proposes a new methodology to analyse the bending fatigue behaviour of notched test specimens. From these tests, it was possible to verify that, despite carrying out the tests with load control, the presence of fibres extends the fatigue life of the concrete after cracking. This effect is of great importance since during the extra lifetime with the cracked concrete, the damage to the concrete will be evident and the corresponding maintenance measures can be carried out. Regarding the analysis of the results, in addition to obtaining a traditional S-N curve, two new criteria have been applied, namely energy and notch growth. From these two new approaches, it was possible to determine critical energy values that can be used as predictive indicators of the collapse of the element. Moreover, from the notch growth analysis, it was possible to determine crack growth rate as a function of the stress conditions for the concrete and the specific geometry. From the comparison among the results obtained from the different tests, a limit cracking index of 0.05 mm can be defined.

## 1. Introduction

Concrete combined with other materials to create hybrid structures has been shown to have given good results in the behaviour of concrete, like the concrete-filled fibre-reinforced polymer, steel composite tube column or filled glass-fibre-reinforced polymer [1,2,3]. Fibre-reinforced concrete includes short, discrete fibres in its composition, randomly and homogeneously distributed in its mass [4]. This concrete constitutes one of the most relevant innovations in the field of special concretes [5]. The use of fibres has increased in the last 40 years, many types of fibres having been successfully adapted to different concrete applications, due to their ability to improve the capabilities of concrete, increasing its use in certain industries [6]. One of the advantages of the incorporation of fibres is in the control of cracking [7,8,9], providing a significant increase in the energy absorption capacity of concrete, even doubling it compared to traditional concrete, and preventing fragmentation at breakage, unlike concrete without fibres [7,10]. In addition, various authors have quantified the improvements that fibres contribute to residual flexural strength [7,8,9,11,12,13,14,15,16]. On the other hand, fibres not only improve the mechanical properties, but they can also contribute to lowering thermal conductivity of concrete too. Several authors have studied this effect, showing that that addition of Polypropylene fibres reduce it for densities lower than 1800 kg/m^3^, having the opposite effect with bigger densities [17,18].

There are different types of fibres according to the material used, but the most usual ones are steel fibres. Regarding the different shapes that these fibres can take, numerous studies have recommended hooked-end ones, since the adhesion between the cement matrix and the fibre is improved, compared to straight or corrugated fibres [16,19].

Although concrete is one of the most commonly used materials in construction, it is not usual to analyse the effect of applying cyclical loading to it. Historically, concrete structures are not commonly subjected to these fatigue loads [20,21,22,23,24]. However, due to the great evolution that the construction sector has undergone in recent decades, which enables the optimization of the designs of structures, the exposure of an element to cyclical loads must be taken into account as it is the most critical efforts for the concrete [21,22,23,25,26], for example in airports pavements or bridge decks. Currently there are several studies that analyse the effect of fatigue on plain concrete using different methodologies including S-N curves, Staircase or Locati methods among others to analyse the fatigue behaviour under compression or bending fatigue of concrete. There are also studies that analyse the effect of fatigue on more special concretes such as recycled concretes. For reinforced concrete, for example Eurocode 2 [27], the design of the concrete structure for fatigue is mainly based on the design of the steel reinforcements for fatigue. On the other hand, there are just a few studies that analyse the effect of fatigue on other fibre-reinforced concrete.

It has been demonstrated that steel fibres contribute to the improvement in flexural, tensile, impact and fatigue strength of concrete mixtures [9,12,28,29]. Specifically, there is a recognized improvement provided by the incorporation of short-range reinforcements such as steel fibres [30] or polypropylene fibres [31,32]. The use of this kind of fibres has been shown to improve bending strength by up to between 25% and 50%. Moreover, the increase in the volume of fibres improves the mechanical properties [28] and increases the number of cycles that the concrete resists [33,34]. The fatigue damage is 1–2 times higher than the damage obtained in a static test in concrete with fibres [35].

Fibres give to the concrete better behaviour under fatigue. It has been verified that the benefit of the addition of fibres to concrete is more pronounced at higher fatigue loading levels and is insignificant at lower levers [35,36]. The number and the presence of fibres influence the load at which the first cracking appears. It has been demonstrated that the fatigue damage is influenced by the type and dimensions of the fibres [37]. Hooked-end fibres display poorer ductility than corrugated ones and the load after breaking decreases faster with corrugated fibres [29]. Some authors [38,39] concluded that the predominant mode of failure is by pulling out of the fibres from the concrete matrix, when the contact between matrix and fibre is reduced. Z. Jun at al. [40] have concluded that there is a relation between the fatigue behaviour and the cyclic relation stress-crack width and that an optimum performance of the fatigue can be achieved optimizing the cyclic stress-crack width relationship. D. Caresso et al. have suggested that the reduction of the ductility is due to lower load fatigue levels [39]. It has been demonstrated too that there is a critical CMOD point that is larger at higher fatigue loads due to the larger amount of energy accumulated in a short period of time in concrete with fibres [36] and the applied load influences the CMOD development, growing as it is increased [39].

Currently, knowledge about the fatigue process and damage mechanisms that occur in concrete is very scarce and even more so in the case of concrete with fibres, due probably to the time consuming for the fatigue tests. In addition, the relation between the cracking and the energy has not been practically studied and parameters that indicates premature failure and of fibre reinforced concrete are needed. In this paper, a complete characterization is done of notched-concrete specimens with commercial hooked-end fibres HE++90/60 under bending fatigue. During this characterization, consisting of 12 tests, the load evolution, actuator displacement and notch opening were recorded continuously throughout the test. From these data, it was possible to determine the fatigue life of the concrete as a function of the stress range applied. Moreover, two new analysis procedures were applied. From one of them, it was possible to determine an energy threshold after which concrete cracking occurs.

## 2. Materials and Methods

### 2.1. Materials

For the manufacture of the concrete, a CEM I 52.5N type cement was used, according to EN 197-1:2011 [41], with a water/cement (w/c) ratio of 0.42. The density of the cement, obtained according to UNE 80103:2013 [42], was 3.12 g/cm^3^.

The aggregates used to produce the concrete were limestone gravel (min/max of 4/12 and 10/20 mm) and limestone sand (min/max of 0/2 and 0/4 mm). The aggregates’ physical properties can be seen in Table 1 and Figure 1 shows the grading of the aggregates, obtained according to EN 933-1 [43].

The selected steel fibres were hooked-end type, HE 90/60 made by ArcelorMittal, with a length (*lf*) of 60 mm, a diameter (*df*) of 0.90 mm and an aspect ratio (*λ = lf/df)* of 67 and a tensile strength of approximately 1200 N/mm^2^. An example of the fibres used is shown in Figure 2.

The concrete mix proportions were obtained by the Fuller method, adding a 0.44% by volume of steel fibres, and are shown in Table 2. The w/c ratio was 0.42 and the superplasticizer additive, MasterEase 5025, was 1% wt. of cement.

### 2.2. Methods

#### 2.2.1. Conventional Mechanical Properties

Compressive strength was determinate, according to EN 83507:2004 [44]. Three different specimens were tested at 28 days. The specimens were cubic of 150 × 150 × 150 mm^3^. All the compressive-tested specimens were cured in a humidity chamber under controlled conditions (20 ± 2 °C and 95 ± 5% humidity).

Flexural tensile strength was determined according to UNE-EN 14651:2007 [45]. Three different specimens were tested at 28 days. The specimens were prismatic of 600 × 150 × 150 mm^3^ and they were cured in a humidity chamber like the compressive ones. The Crack Mouth Opening displacement (*CMOD*) was calculated from the recorded vertical displacement (*ρ*), according to the standard [Equation (1)].
(1)$$CMOD=\frac{\rho -0.04}{0.85}$$

#### 2.2.2. Fatigue Tests

The fatigue tests were performed at 90 days, using the Staircase method [46,47]. Twelve specimens of 600 × 150 × 150 mm^3^ were tested. All the fatigue-tested specimens were cured in a humidity chamber under controlled conditions (20 ± 2 °C and 95 ± 5% humidity).

To prepare the specimens for the test, a crack was made, based on the standard EN 14651:2007 [45]. The cracks were cut with a wet disc saw and had a width of 5 mm and a height of 25 ± 2 mm, traversing the width of the specimen, as is shown at Figure 3.

To measure the CMOD a COD extensometer was used. Two metal pieces were fixed to the two sides of the crack, in the centre of it, and an extensometer was fitted, taking the specimens to breakage and recording the Load-COD data.

The specimen was placed on a cylinder of 30 mm diameter, on the opposite side to the cracked face. On the cracked face, two cylinders were place at a 500 mm separation, with the same dimensions as the first one. The test assembly of the specimen can be seen in Figure 4.

Fatigue testing begins by loading the specimen to the medium load level at a speed of 0.5 MPa/s. When this mean load value has been reached, sinusoidal loads are applied between the maximum and minimum load values indicated in Table 3 at a frequency of 15 Hz. Different load scenarios were used to analyse the fatigue behaviour of fibre-reinforced concrete. The minimum load applied in all the tests was 2 kN and the maximum was varied to compare different states. In this Table 3 the values of stress range (Δσ) and stress ratio (R) for each test are also indicated. The stress range corresponding to the applied load were obtained using Equation (2) taken from EN-14651 which considers the geometry of each of the specimens.
(2)$$\sigma =\frac{3\times F\times L}{2\times b\times h^{2}}$$
where *σ* = stress; *F* = load applied; *L* = distance between the two lower cylinders; *b* = specimen width; *h* = distance from the base to the beginning of the notch.

During all the fatigue tests, the testing time, load, actuator displacement (ACT) and COD aperture of two cycles of every 100 were registered during the whole test. To analyse these data, and due to the waveform, a regression to a sinusoidal wave was done by mean of a one degree Fourier regression, see Equation (3), which was transformed into Equation (4) by means of Equation (5) [48]. This one-degree Fourier regression was applied using Matlab software [49]. In Figure 5 an example of the comparison of the raw data and the regression can be seen. After this regression, the time variation of all the parameters which define a sinusoidal wave could be analysed, such as mean value or amplitude.
(3)$$y\left(t\right)=a_{0}+a_{1}\times \cos\left(w\times t\right)+b_{1}\times \sin\left(w\times t\right)$$
(4)$$y_{t}=y_{0}+A\times sen\left(w\times t+\varphi_{0}\right)$$
(5)$$y_{0}=a_{0}\;;\;A=\sqrt{a_{1}^{2}+b_{1}^{2}}\;;\;\varphi_{0}=\arctan(\frac{a_{1}}{b_{1}})$$

In addition to analysing the effect of the cycle concatenation on the mean value and amplitude, the effect on the area of the loop generated by drawing the COD opening versus the load was also analysed. Two different types of area were analysed, on the one hand, the area representing the energy provided Figure 6 (left) and, on the other, the stored energy Figure 6 (middle).

The total energy (stored + dissipated) represents the energy provided to the system to produce the deflection of the specimen. This total energy is divided into two types of energy, stored energy, and dissipated energy. The dissipated energy is the energy released by the specimen during the cycle. On the other hand, the stored energy is the energy that the material return/restore cycle by cycle.

During the analysis of the data, it was found that in all cases there were two clearly differentiated behavioural zones, before and after the cracking of the concrete. Figure 7 shows the evolution of each of the parameters analysed throughout the test. The applied load wave is constant throughout the test, before and after the cracking. This is because the test is performed under load control. In the case of the evolution of the ACT and the COD, it is possible to clearly identify the moment of concrete cracking. From that point onwards, it can be seen how the values of both parameters begin to evolve rapidly, especially in the case of COD. In the case of the evolution of the curve load vs. COD, the moment of the cracking can also be easily appreciated. From the cracking onward the area begins to grow rapidly and the curve slopes to the right. Once the results of the different tests were compared, it was concluded that until the cracking of the concrete the results were quite homogenous but, when the concrete cracks and the fibres are responsible for the specimen behaviour, the results become heterogeneous due to the random distribution of the fibres [50] and because fatigue is a phenomenon that focuses on a specific point of the sample, the crack. For this reason, the analysis was focused on the behaviour before cracking.

Another additional parameter was analysed during the fatigue tests, the COD opening velocity. This velocity was obtained as the derivative of the COD opening as a function of time. This curve has a U shape, see Figure 8, since the initial part and the final part are phenomena of the beginning of the test or near to the crack. To compare the effect of the stress range of the test on the growth velocity of the COD opening, a value in the central zone of the stationary part was averaged, see Figure 8.

## 3. Results and Discussion

### 3.1. Mechanical Properties

The average compressive strength at 28 days is 50 MPa. In Table 4, the results of the compressive strength test in the 3 specimens can be seen.

Table 5 shows the residual flexural strength for the 2.5 mm crack opening (f_R,3_, COD = 2.5 mm). The average of these values is f_R,3_ = 6.27 MPa. In Figure 9 a compressive and a flexural specimen can be seen.

### 3.2. S-N Curve

Figure 10 shows the number of cycles to either cracking (red circles) or breakage (blue squares) depending on the stress range of the test.

The cycles to cracking of the specimens fit very well with an S-N curve. On the other hand, the number of cycles until specimen breakage does not fit so well. This is because, after cracking of the concrete, the fibres are the ones that manage the fatigue process. For this reason, the orientation and number of fibres holding the crack will define the behaviour of the concrete. This result is in agreement with the results obtained by Jose Rios and Héctor Cifuentes [51], who concluded that the presence of fibres increased the scatter of the results during the flexural fatigue of fibre-reinforced concretes. Figure 11 shows examples of cracking during the fatigue tests.

The fatigue life after cracking is very variable and does not seem to follow a clear trend. In all cases where component failure is reached, the presence of fibres has increased the life of the component to some extent. This does not have to happen because it is a concrete with fibres, since, the tests generally done on concrete with fibres are carried out under machine displacement control, while in this case they were done under load control.

The additional cycles provided by the presence of fibres, although generally not many, make the difference between having a fragile break. A fragile break means no apparent signs that the break will occur and having clear signals that the break is nearby. This, in a real situation, will enable measures to be taken after breakage becomes evident.

### 3.3. Crack Opening during the Fatigue Tests

Figure 12 shows the parameters recorded during the test, specifically a sample of the results of test 9. Figure 12a shows the appearance of these parameters at the beginning of the test, while Figure 12b shows the appearance of these parameters at the end of the test. The load values remain constant, which is because these tests were carried out under load control. About the displacement of the ACT, it increased from a displacement of 0.3 mm to 0.6 mm. The opening of the COD is the parameter whose behaviour varies most. At the beginning of the test the crack barely opened 8 microns, while in the final part of the test, with the specimen already cracked, the crack opened more than 0.3 mm.

Figure 13 shows the evolution of the maximal crack opening during the fatigue tests. Figure 13 shows different curves, representing different test specimens with different stress range values. These curves all have a similar shape. They are initially approximately linear and stable but then in all cases increase approximately exponentially. It can also be seen that the higher the stress ratio applied during the fatigue test, the higher the curves are and the further to the left. This means that the larger the crack opening, the shorter the duration of the test. From the comparison among the results obtained from the different tests, a limit cracking index of 0.05 mm can be defined. From this observation it could be assumed that when the crack opening increases beyond this value of 0.05 mm, concrete cracking occurs.

As the specimen is subjected to a fatigue test based on the application of sinusoidal loads, the specimen is simultaneously subjected to two types of load. On the one hand, a constant load equivalent to the average value of the wave applied and, on the other hand, a sinusoidal load ranging from a maximum load *F* to a minimum of *−F*. Each of these two loads will have a different impact on the damage caused to the concrete. The constant applied load will accumulate creep damage, while in the case of cyclic loading, it will accumulate fatigue damage. Figure 14 analyses each of these effects separately. Figure 14 shows the evolution of the average crack opening value over the course of the test. Figure 14 shows the evolution of the crack opening amplitude as a function of the number of cycles.

It can be seen that Figure 14 has a very similar shape to Figure 13, while in the case of Figure 14, an appreciable difference can be observed in the shape of the curves. In the case of the evolution of the mean value, an evolution can be seen from the first cycles in practically all cases. In the case of the amplitude, this is a much more stable parameter which is practically only appreciable in the last cycles before cracking occurs. From these results, it can be assumed that in tests where the load values are sufficiently low, the damage is predominantly creep damage, until a limit is reached that allows fatigue damage to begin to increase.

Figure 15 shows the opening velocity of COD in the stationary section as a function of the stress range. It should be remembered that the cases were analysed only before the concrete cracking occurred. For this reason, for example Test 6 could not be included since the cracking occurs so early that a stationary crack growth rate is not reached before cracking.

Increasing the stress range value increases the growth rate of the maximum opening value, the average value, and the amplitude. Based on the analysis of the evolution of the growth of the notches throughout the test, it could be deduced that, in the case that the stress range is small, the evolution of the opening is practically that of the medium level. In the case of a test with high tensional range values, the values of crack opening amplitude and the mean value of it can become similar.

### 3.4. Energy Evolution during the Fatigue Tests

Figure 16 shows the evolution of the provided and stored energy during each fatigue test.

In Figure 16, for higher stress values greater energy is required to apply the load and greater energy is stored by the sample.

Figure 17 shows the growth rate of the energy supplied in the stationary zone as a function of the stress range of the test.

From Figure 18, there is a relationship between the velocity of provision of energy of provided energy and the stress range of the test. In addition, as previously indicated, there is a critical energy value that indicates that the test specimen will break, so it could be used as a parameter to determine the remaining life of the concrete. However, it should be noted that since it is an energy parameter, it is specific to each geometry.

## 4. Conclusions

The following conclusions can be drawn from the work:An S-N curve of the material was obtained before it was cracked. Once the concrete has cracked, the quantity and orientation of the fibres is so important in the response of the concrete to cyclic loads that it leads to extremely heterogeneous behaviour.The presence of fibres was proven to increases the fatigue life of concrete, even in tests carried out under load control. This provides a time between the concrete breakage and the structural breakage, which can be used to take appropriate corrective measures.A critical crack opening was detected after which concrete cracking occurs, which could be used as an indicator the premature failure of the specimen.A critical energy value was observed that indicates that the test specimen will break, so it could be used as a parameter to determine the remaining life of the concrete. However, it should be noted that since it is an energy parameter, it is specific to each geometry.It was verified that the mechanism producing failure of the concrete is caused by microcracks inherent to concrete in the notch beginning to grow due to the loads applied. The velocity of growth of the cracks depends on the range of stress applied in the test, a correlation existing between them. However, geometry probably also has an influence.

There are not many research on the fatigue behavior of fiber reinforced. However, the importance that this material may have soon, thanks to its better performance and savings in execution, necessary to investigate on it. Hence the importance of the results presented and the obtained results, opening the door to a new type of analysis regarding behaviour of concrete under bending fatigue. Although it is important to emphasize that this study has to be complemented with future studies in which not only different specimen geometries are analysed, but also different notch sizes.

## Figures and Tables

**Figure 1 materials-14-05854-f001:**
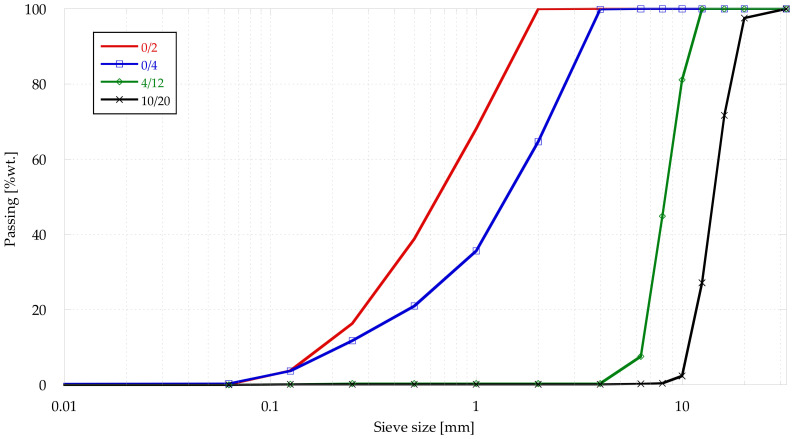
Grading curve.

**Figure 2 materials-14-05854-f002:**
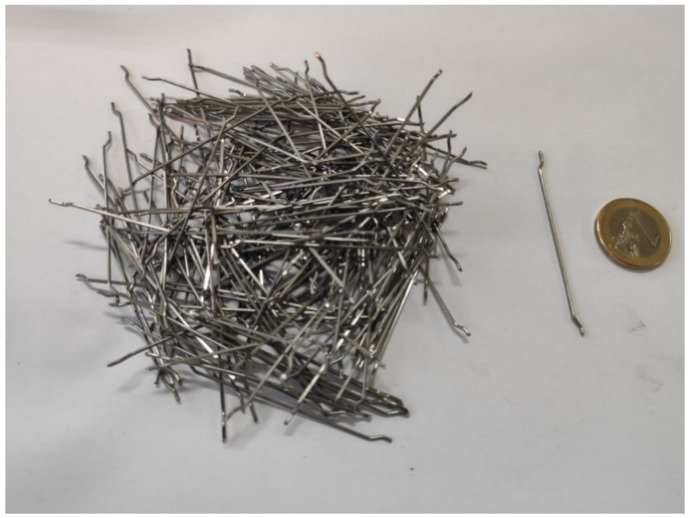
Hooked-end steel fibres used.

**Figure 3 materials-14-05854-f003:**
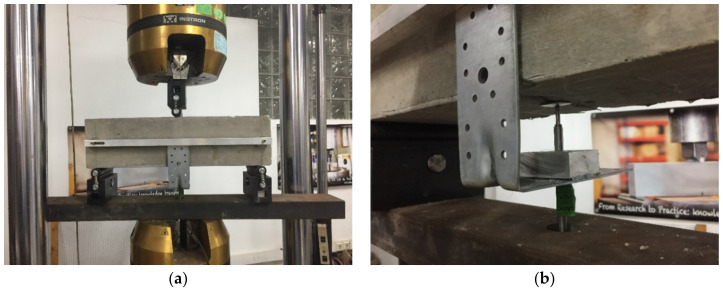
Flexural tensile strength test (**a**) and measurement of vertical displacement with a LVDT (**b**).

**Figure 4 materials-14-05854-f004:**
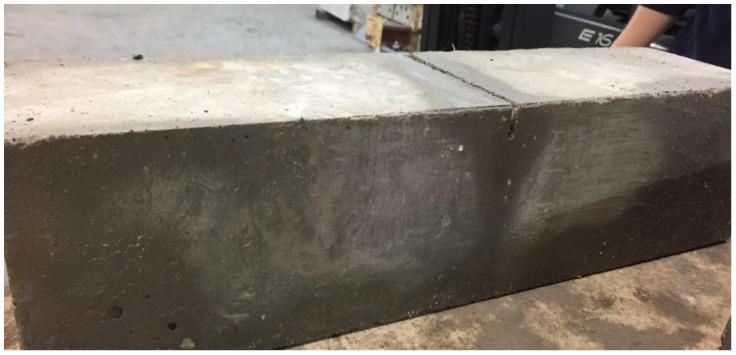
Fatigue specimen.

**Figure 5 materials-14-05854-f005:**
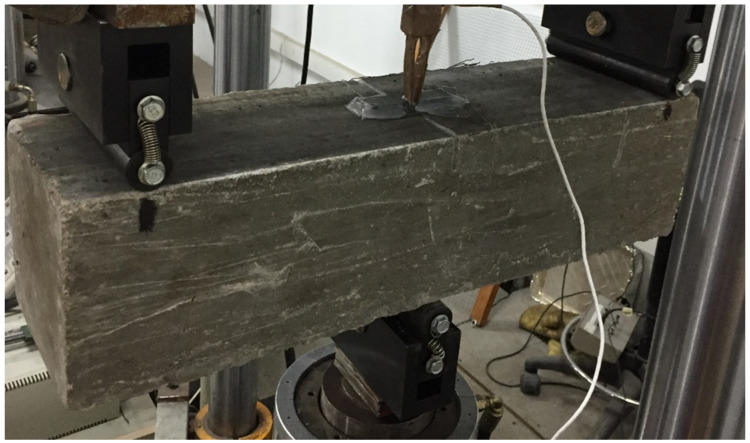
Fatigue test.

**Figure 6 materials-14-05854-f006:**
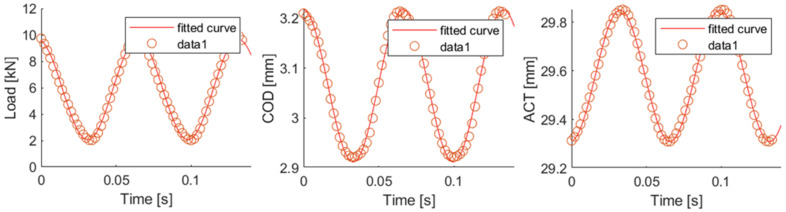
Data fitting.

**Figure 7 materials-14-05854-f007:**
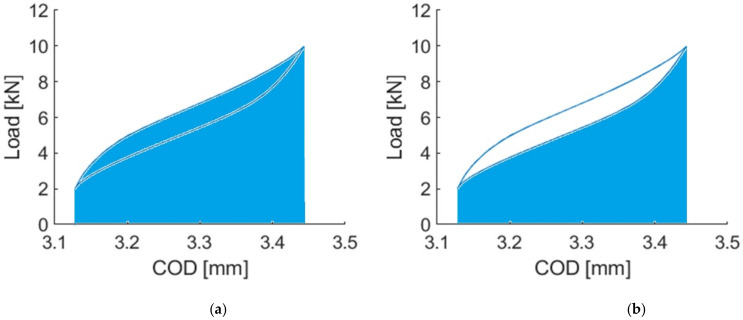
Determining the energy provided (**a**) and stored (**b**).

**Figure 8 materials-14-05854-f008:**
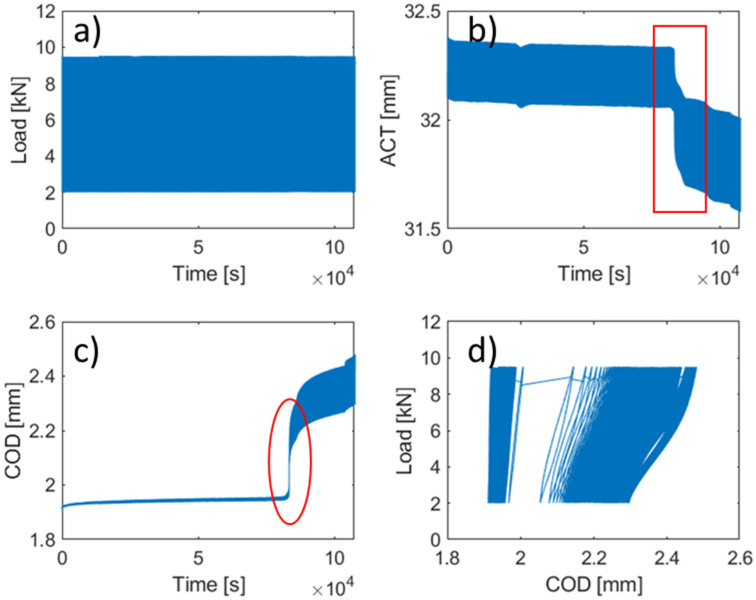
Identification of two clearly differentiated phases within the test, before and after cracking (figure corresponding to test 10). Evolution of load measurement during the fatigue test (**a**). Evolution of actuator displacement during the fatigue test (**b**). Evolution of COD measurement during the fatigue test (**c**). Evolution of the curve COD vs. Load during the fatigue test (**d**).

**Figure 9 materials-14-05854-f009:**
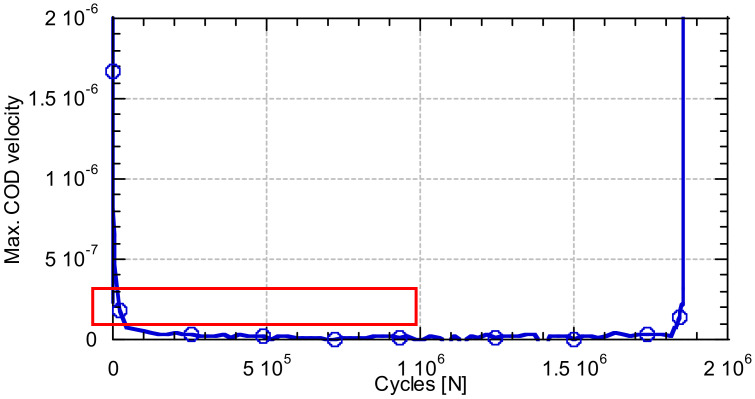
Evolution of the velocity of the parameters analysed (example corresponding to test 2).

**Figure 10 materials-14-05854-f010:**
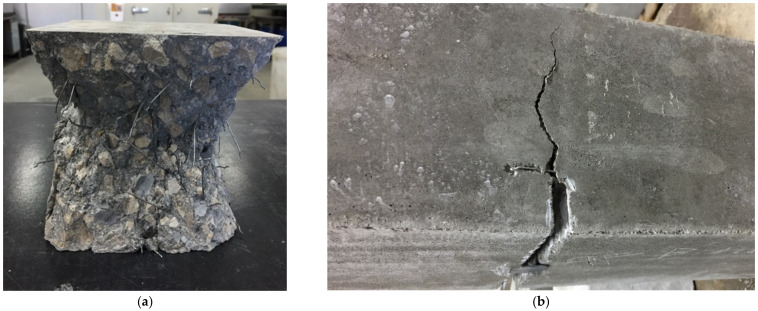
Specimens after compressive test (**a**) and flexural test (**b**).

**Figure 11 materials-14-05854-f011:**
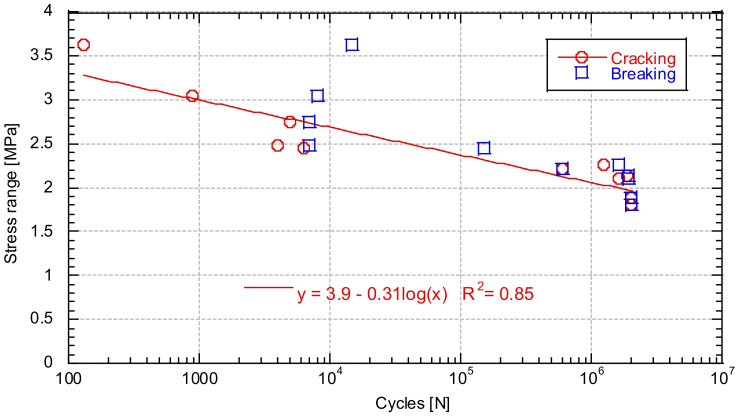
S-N curve.

**Figure 12 materials-14-05854-f012:**
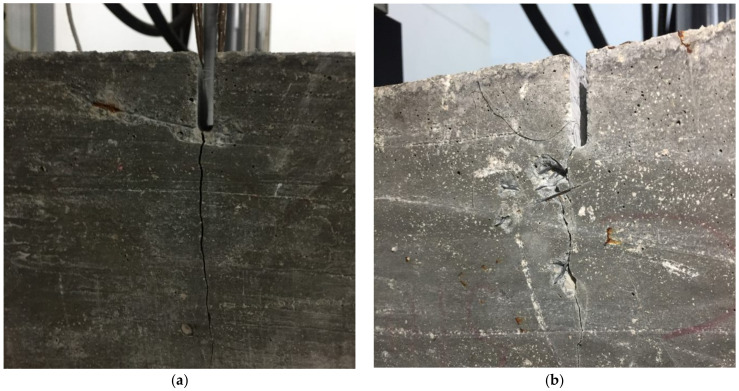
Cracking of concrete in the fatigue test specimen side 1 (**a**) and specimen side 2 (**b**).

**Figure 13 materials-14-05854-f013:**
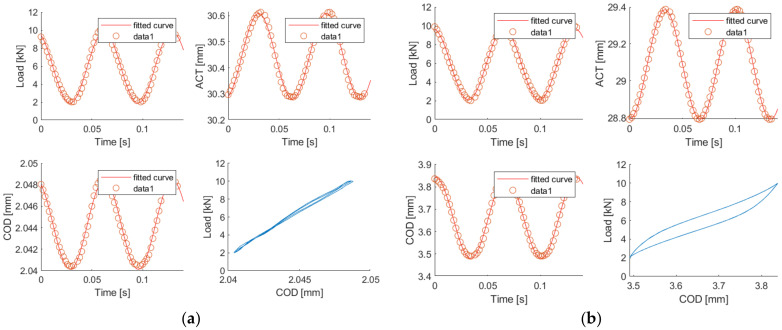
Comparison of the specimen response during the fatigue test. In an uncracked specimen (**a**). In a cracked specimen (**b**).

**Figure 14 materials-14-05854-f014:**
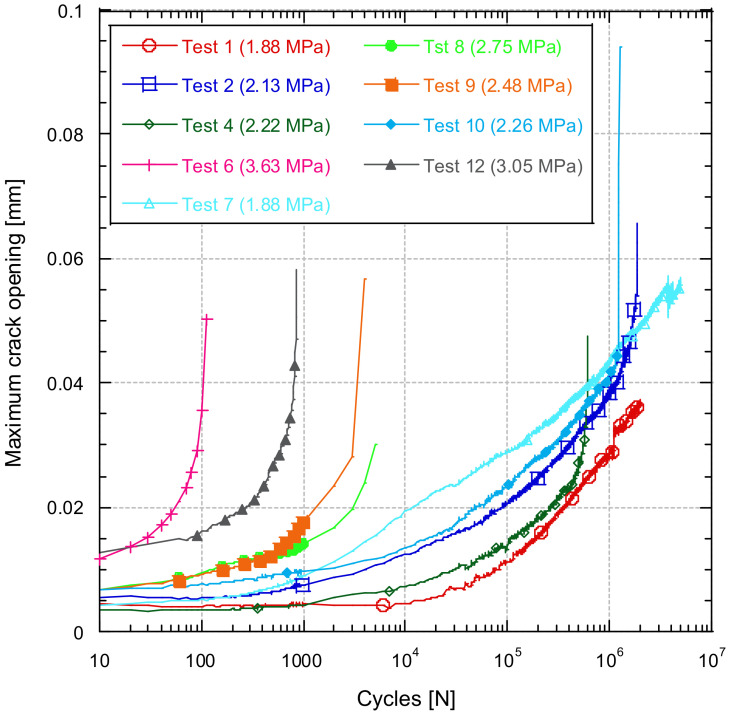
Evolution of the crack maximum opening as a function of the number of cycles.

**Figure 15 materials-14-05854-f015:**
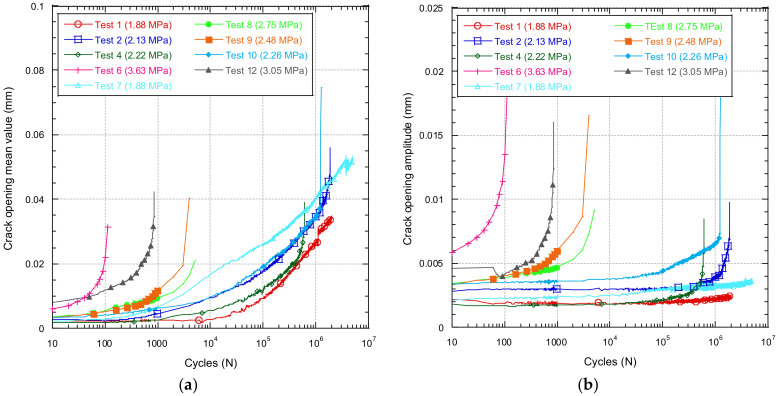
Evolution of the mean crack opening value (**a**) and amplitude (**b**) as a function of the number of cycles.

**Figure 16 materials-14-05854-f016:**
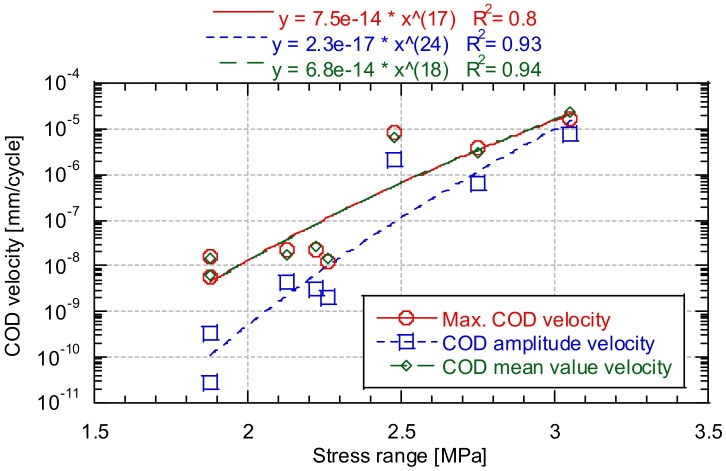
COD velocity as a function of the test stress range.

**Figure 17 materials-14-05854-f017:**
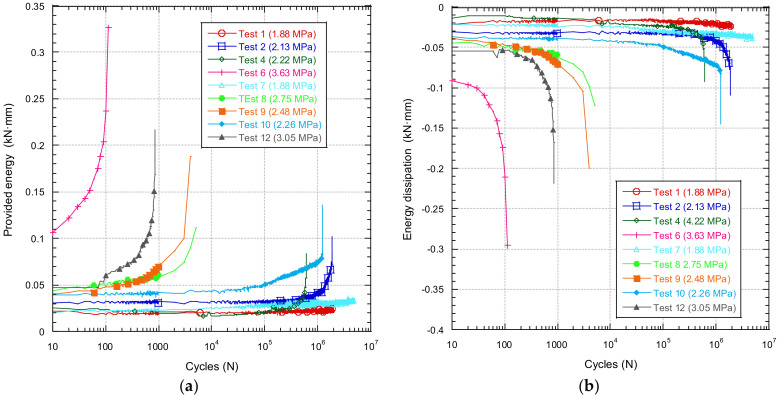
Evolution of the provided (**a**) and stored (**b**) energy during the fatigue test.

**Figure 18 materials-14-05854-f018:**
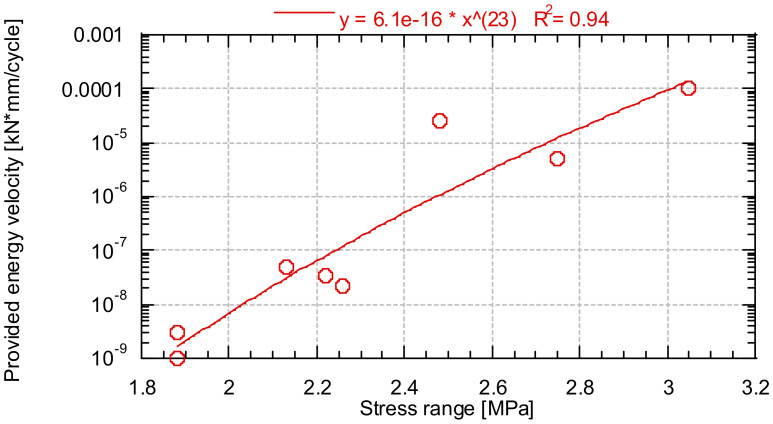
Provided energy velocity as a function of the test stress range.

**Table 1 materials-14-05854-t001:** Aggregate physical properties.

Size [mm]	Sand Equivalent	Absorption [%]	Density [g/cm^3^]
0/2	>75	0.49	2.69
0/4	>80	0.49	2.69
4/12	-	0.54	2.70
10/20	-	0.54	2.68

**Table 2 materials-14-05854-t002:** Mix proportions.

Material	Mix [kg/m^3^]
0/2	480
0/4	480
4/12	480
10/20	480
Cement	390
Water	165
Additive	3.9
Fibres	35

**Table 3 materials-14-05854-t003:** Test parameters.

Specimen	Fmin [kN]	Fmax [kN]	R	b [mm]	h [mm]	Δσ [MPa]
F-01	2	8	0.25	153.6	124.8	1.881
F-02	2	9	0.22	154.6	126.2	2.132
F-03	2	8	0.25	153.0	127.6	1.806
F-04	2	9	0.22	156.4	123.0	2.219
F-05	2	10	0.20	162.0	124.0	2.109
F-06	2	15	0.13	159.0	130.0	3.628
F-07	2	8.5	0.24	163.5	125.8	1.884
F-08	2	11	0.18	157.0	125.0	2.752
F-09	2	10	0.20	160.0	123.0	2.479
F-10	2	9.5	0.21	157.0	126.0	2.257
F-11	2	9.8	0.20	153.0	125.0	2.447
F-12	2	12	0.17	160.0	124.0	3.049

**Table 4 materials-14-05854-t004:** Compressive strength results at 28 days.

Specimen Code	Compressive Strength [MPa]
Specimen C1	53.80
Specimen C2	51.28
Specimen C3	46.48

**Table 5 materials-14-05854-t005:** Flexural tensile strength results at 28 days.

Specimen Code	f_R,3_ [MPa]
Specimen F1	5.80
Specimen F2	6.83
Specimen F3	6.17

## Data Availability

Not applicable.

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
