# Peer review of "Effect of Steel Fibre Reinforcement on Flexural Fatigue Behaviour of Notched Structural Concrete"

_materials, 2021, doi:10.3390/ma14195854_

Round 1

Reviewer 1 Report

This paper proposes a new methodology to analyse the bending fatigue behaviour of notched test specimens. They find that the presence of fibres extends the fatigue life of the concrete after cracking. This effect is important because during the extra lifetime with the cracked concrete, the damage to the concrete will be evident and the corresponding maintenance measures can be carried out. In addition to obtaining a traditional S-N curve, two new criteria have been applied, namely energy and notch growth. The results obtained open the door to a new type of analysis regarding behaviour of concrete under bending fatigue, although it is important to emphasize that this study has to be complemented with future studies in which not only different specimen geometries are analysed, but also different notch sizes.

All in all, the amount of experiments is relatively sufficient, and relatively rich experimental results have been obtained, and more practical reference suggestions have been put forward. Generally speaking, the topic selection is relatively new, the overall academic level is high, but the logic is general, and a relatively practical conclusion has been obtained. There are still many problems, I hope the author will carefully modify the description:

  1. Line 100,“The aggregates used to produce the concrete were limestone gravel (4/12, 10/20)”,What do you mean about “(10/20)"?
  2. Line 114 ,What are the units in Table 2?
  3. Line 126, Is there a difference between COD and CMOD?
  4. Line 157, What are the reasons for selecting the minimum and maximum loads in Table 3? Is there a control group for each experimental group?  The stress in formula 2 is calculated based on which load?  Why is there no gradient in the stress setting?  Have you considered the stress ratio?
  5. Line 176, Please explain in detail the specific areas and meanings of the energy provided (a) and stored (b) in Fig. 6.
  6. Line 181, What do you mean about ACT?
  7. Line 248,Is there a difference between COD, CMOD and DOC?
  8. Line 252 and line 254,The title of Figure 13 appears twice and is inconsistent.
  9. Supplementary explanation of the details of the fatigue test, such as initial loading rate, maximum and minimum load ratio, loading mode, loading frequency and other experimental parameters

Author Response

Reviewer #1 (changes in blue)

This paper proposes a new methodology to analyse the bending fatigue behaviour of notched test specimens. They find that the presence of fibres extends the fatigue life of the concrete after cracking. This effect is important because during the extra lifetime with the cracked concrete, the damage to the concrete will be evident and the corresponding maintenance measures can be carried out. In addition to obtaining a traditional S-N curve, two new criteria have been applied, namely energy and notch growth. The results obtained open the door to a new type of analysis regarding behaviour of concrete under bending fatigue, although it is important to emphasize that this study has to be complemented with future studies in which not only different specimen geometries are analysed, but also different notch sizes.

All in all, the amount of experiments is relatively sufficient, and relatively rich experimental results have been obtained, and more practical reference suggestions have been put forward. Generally speaking, the topic selection is relatively new, the overall academic level is high, but the logic is general, and a relatively practical conclusion has been obtained. There are still many problems, I hope the author will carefully modify the description:

Thank you very much for your efforts in reviewing this paper, we appreciate your help in improving this work. We will respond to all your comments below.

  1. Line 100,“The aggregates used to produce the concrete were limestone gravel (4/12, 10/20)”,What do you mean about “(10/20)"?

We mean that the size of these aggregates is between 10 and 20 mm (diameter). In order to clarify this, the sentence has been completed with the following information “gravel (min/max of 4/12 and 10/20 mm) and limestone sand (min/max of 0/2 and 0/4 mm).”

  1. Line 114 ,What are the units in Table 2?

Thank you very much for your comment. The units are kg/m3, but due to an oversight we forgot to add it. Table 2 was modified as follows:

Material

Mix [kg/m3]

0/2

480

0/4

480

4/12

480

10/20

480

Cement

390

Water

165

Additive

3.9

Fibres

35

  1. Line 126, Is there a difference between COD and CMOD?

In general, COD refers to the device used to measure the crack opening, while CMOD (Crack Mouth Opening displacement) is the measured parameter. Thanks to your comment we have noticed that in some cases (e.g. line 126) these two terms have been mixed up. The difference between the COD and the CMOD was explained in lines 141 to 143 “To measure the CMOD a COD extensometer was used. Two metal pieces were fixed to the two sides of the crack, in the centre of it, and an extensometer was fitted, taking the specimens to breakage and recording the Load-COD data.” Also, the whole document has been checked for those cases where by mistake these two terms have been mixed up.

  1. Line 157, What are the reasons for selecting the minimum and maximum loads in Table 3? Is there a control group for each experimental group? The stress in formula 2 is calculated based on which load? Why is there no gradient in the stress setting? Have you considered the stress ratio?

In this type of test, the specimen is always subjected to a load that pushes it. This implies that the theoretical minimum load would be 0 kN. From a practical point of view, in a test where it is only possible to apply compressive loading on the specimen, it is not feasible to ask the test machine to find the "0 kN" load, it is necessary to set a preload as a minimum value. In concrete fatigue testing there are mainly two options, on the one hand to use a constant minimum value or a fixed stress ratio (in concrete usually 0.1). In our case, we chose to use a constant minimum strength value in order to be able to compare the evolution of the parameters under the minimum load value. In this case, it was decided to set the minimum load at 2 kN, which corresponds to approximately 5% of the static breaking value. Regarding the maximum load, it was modified in order to fully define the S-N curve.

Formula 2 is taken from EN-14651 where the flexural strength of fibre-reinforced concrete specimens in the presence of a notch is determined. Although the presence of a notch can cause stress gradients, since there is a standard that supports the validity of this formula, it was decided to use it. Equation 2 is used to transform any force into the maximum stress generated at the centre of specimen at the height of the fibre where the notch begins. But in the case of fatigue, its main interest is in the analysis of the stress variation, corresponding to the force variation.

  1. Line 176, Please explain in detail the specific areas and meanings of the energy provided (a) and stored (b) in Fig. 6.

Indeed, these are two very important terms that need to be clarified. The following text was included in lines 180 to 184: “The total energy (stored+dissipated) represents the energy provided to the system to produce the deflection of the specimen. This total energy is divided into two types of energy, stored energy, and dissipated energy. The dissipated energy is the energy released by the specimen during the cycle. On the other hand, the stored energy is the energy that the material return/restore cycle by cycle.”.

  1. Line 181, What do you mean about ACT?

The term ACT refers to the displacement of the actuator applying the load, i.e. the deflection of the specimen when the load is applied. This is indeed a parameter that has to be indicated in the text, so the following text was added, line 159 “During all the fatigue tests, the testing time, load, actuator displacement (ACT)”.

  1. Line 248,Is there a difference between COD, CMOD and DOC?

As explained in question 2, there is a slight difference between COD and CMOD although the CMOD value corresponds to the COD measurement. Regarding the DOC, this is a typo. The whole document has been revised to solve these problems.

  1. Line 252 and line 254,The title of Figure 13 appears twice and is inconsistent.

Indeed, this part of the text can be confusing. Regarding the title of the figure appearing twice, this is not the case, it is a sentence indicating that it is shown in the figure and the caption of the figure. But they are indeed inconsistent. To solve both problems, the following solution was chosen: The text “Fig. 13 shows the evolution of the maximal crack opening during the fatigue tests.” Was moved back to the figure and joined with the next paragraph in order to clarify that it is not a title. Also, the caption of the figure was modified specifying that the parameter represented is the maximum crack opening.

  1. Supplementary explanation of the details of the fatigue test, such as initial loading rate, maximum and minimum load ratio, loading mode, loading frequency and other experimental parameters

The authors agree that the fatigue tests needed to be better explained, so the text was rewritten as follows and a column with the stress ratio was added to Table 3: lines 150 to 158: “Fatigue testing begins by loading the specimen to the medium load level at a speed of 0.5 MPa/s. When this mean load value has been reached, sinusoidal loads are applied be-tween the maximum and minimum load values indicated in Table 3 at a frequency of 15 Hz. Different load scenarios were used to analyse the fatigue behaviour of fibre-reinforced concrete. The minimum load applied in all the tests was 2 kN and the maximum was varied to compare different states. In this Table 3 the values of stress range (Δσ) and stress ratio (R) for each test are also indicated. The stress range corresponding to the applied load were obtained using Equation 2 taken from EN-14651 which considers the geometry of each of the specimens.”.

Reviewer 2 Report

Minor revision

This manuscript introduces that the paper proposes a new methodology to analyse the bending fatigue behaviour of notched test specimens. From these tests, it was possible to verify that, in spite of carrying out the tests with load control, the presence of fibers extends the fatigue life of the concrete after cracking. In summary, the research is interesting and provides valuable results, but the current document has several weaknesses that must be strengthened in order to obtain a documentary result that is equal to the value of the publication.

Title, Abstract and Keywords:

  • The abstract is complete and well-structured and explains the contents of the document very well. Nonetheless, the part relating to the results could provide numerical indicators obtained in the research.

Chapter 1: Introduction

  • The first paragraph introducing the research topic gives a too simple, and even incomplete, view of the problems related to your topicand should be revised and completed with citations to authority references (Axial compression behavior of recycled-aggregate-concrete-filled GFRP–steel composite tube columns).
  • The novelty of the study is not apparentenough. In the introduction section, please highlight the contribution of your work by placing it in context with the work that has done previously in the same domain.
  • On a general level, the study of the proposeddetection techniques is reasonable, and the explanation of the objectives of the work may be valid. However, the limitations of your work are not rigorously assumed and justified.

Chapter 3: Results and Discussion

  • There are a large number of English long sentences in this chapter. It is recommended to modify it to multiple simple sentences to help readers understand.

Chapter 4:  Conclusions

  • The author did not mention the application prospects about the actual engineering in this chapter, and the authors explained in detail.

Author Response

Reviewer #2: (Changes in green)

This manuscript introduces that the paper proposes a new methodology to analyse the bending fatigue behaviour of notched test specimens. From these tests, it was possible to verify that, in spite of carrying out the tests with load control, the presence of fibers extends the fatigue life of the concrete after cracking. In summary, the research is interesting and provides valuable results, but the current document has several weaknesses that must be strengthened in order to obtain a documentary result that is equal to the value of the publication.

Thank you very much for your efforts in reviewing this paper, we appreciate your help in improving this work. We will respond to all your comments below.

  1. Title, Abstract and Keywords:

The abstract is complete and well-structured and explains the contents of the document very well. Nonetheless, the part relating to the results could provide numerical indicators obtained in the research.

Thank you for your appreciations, add it the abstract, with this sentence: “From the comparison among the results obtained from the different tests, a limit cracking index of 0.05 mm can be defined.”

  1. Introduction

The first paragraph introducing the research topic gives a too simple, and even incomplete, view of the problems related to your topic and should be revised and completed with citations to authority references (Axial compression behavior of recycled-aggregate-concrete-filled GFRP–steel composite tube columns).

Thank you for your appreciations, added in the article, with this sentence:

“Concrete combined with other materials to create hybrid structures has been shown to have given good results in the behaviour of concrete, like the concrete-filled fibre-reinforced polymer, steel composite tube column or filled glass-fibre-reinforced polymer”

The novelty of the study is not apparent enough. In the introduction section, please highlight the contribution of your work by placing it in context with the work that has done previously in the same domain.

We added this sentence:

“In addition, the relation between the cracking and the energy has not been practically studied and parameters that indicates premature failure and of fibre reinforced concrete are needed”

On a general level, the study of the proposed detection techniques is reasonable, and the explanation of the objectives of the work may be valid. However, the limitations of your work are not rigorously assumed and justified.

In the conclusions section the following sentence has been included “The results obtained open the door to a new type of analysis regarding behaviour of concrete under bending fatigue, although it is important to emphasize that this study has to be complemented with future studies in which not only different specimen geometries are analysed, but also different notch sizes.”

  1. Chapter 3: Results and Discussion

There are a large number of English long sentences in this chapter. It is recommended to modify it to multiple simple sentences to help readers understand.

Indeed, in some parts of the manuscript excessively long sentences were used, making them difficult to understand. The following sentences have been replaced to make them easier to understand:

  • Lines 225 to 228: “This is because, after cracking of the concrete, the fibres are the ones that manage the fatigue process. For this reason, the orientation and number of fibres holding the crack will define the behaviour of the concrete.”.
  • Lines 240 to 424: “The additional cycles provided by the presence of fibres, although generally not many, make the difference between having a fragile break. A fragile break means no ap-parent signs that the break will occur, and having clear signals that the break is nearby.”.
  • Lines 259 to 265: “It can also be seen that the higher the stress ratio applied during the fatigue test, the higher the curves are and the further to the left. This means that the larger the crack opening, the shorter the duration of the test. From the comparison among the results obtained from the different tests, a limit cracking index of 0.05 mm can be defined. From this observation it could be assumed that when the crack opening increases beyond this value of 0.05 mm, concrete cracking occurs.”.
  • Lines 272 to 275: “Fig. 14 analyses each of these effects separately. Fig. 14 (a) shows the evolution of the average crack opening value over the course of the test. Fig. 14 (b) shows the evolution of the crack opening amplitude as a function of the number of cycles.”.
  • Lines278 to 281: “In the case of the evolution of the mean value, an evolution can be seen from the first cycles in practically all cases. In the case of the amplitude, this is a much more stable parameter which is practically only appreciable in the last cycles before cracking occurs.”.
  • Lines 286 to 289: “It should be remembered that the cases were analysed only before the concrete cracking occurred. For this reason, for example Test 6 could not be included since the cracking oc-curs so early that a stationary crack growth rate is not reached before cracking.”.
  1. Chapter 4: Conclusions

The author did not mention the application prospects about the actual engineering in this chapter, and the authors explained in detail.

The conclusions have been completed with the following paragraph: “There are not many research on the fatigue behavior of fiber reinforced. However, the importance that this material may have soon, thanks to its better performance and savings in execution, necessary to investigate on it. Hence the importance of the results presented and the obtained results, opening the door to a new type of analysis regarding behaviour of concrete under bending fatigue. Although it is important to emphasize that this study has to be complemented with future studies in which not only different specimen geometries are analysed, but also different notch sizes.”

Reviewer 3 Report

  1. While providing the benefits of fibres in concrete, the authors have provided in general the benefits, however, there seems that some benefits are not mentioned. Such as the fibre inclusion could contribute to lowering the thermal conductivity of concrete. Please, review the following references with the goal to provide comments and suggestions for future developments:
    1. https://doi.org/10.1088/1757-899X/271/1/012058
    2. https://doi.org/10.1051/matecconf/201815003008 
  2. The steel fibres used in this study was hooked-end type as mentioned in Line 107. Could I ask the authors to provide a picture of the steel fibres for clarity.
  3. How much % by vol. fraction was the steel fibres used?
  4. Eq. 1, CMOD, what does CMOD term stand for? in the text above, there is crack opening (COD) but no mention of CMOD.
  5. For equations, please provide what which term means, like in Eq. 2, σ = 3 * F * L / 2 * b * h^2, please write, "where F = ...., L = .... etc" for more clarity.
  6. I am not sure, if its only which me or not, in Figures 13, 14 and 16, the legend in yellow colour is hard to read, if possible could the authors can the colour from yellow to another.
  7. Please conduct a comprehensive English proofread to remove the grammatical mistakes.

All in all, this article is worth publishing in Materials journal as it fits the scope of the journal and also contributes to the scientific advances.

Author Response

Reviewer #3: (Changes in RED)

Thank you very much for your efforts in reviewing this paper, we appreciate your help in improving this work. We will respond to all your comments below.

  1. While providing the benefits of fibres in concrete, the authors have provided in general the benefits, however, there seems that some benefits are not mentioned. Such as the fibre inclusion could contribute to lowering the thermal conductivity of concrete. Please, review the following references with the goal to provide comments and suggestions for future developments:
  2. https://doi.org/10.1088/1757-899X/271/1/012058
  3. https://doi.org/10.1051/matecconf/201815003008

Thank you, we appreciate your comment and we have implemented this in the paper and the references included:

“On the other hand, fibres not only improve the mechanical properties, but they can also contribute to lowering thermal conductivity of concrete too. Several authors have studied this effect, showing that that addition of Polypropylene fibres reduce it for densities lower than 1800 kg/m3, having the opposite effect with bigger densities [17][18].”

  1. The steel fibres used in this study was hooked-end type as mentioned in Line 107. Could I ask the authors to provide a picture of the steel fibres for clarity.

Thank you, a picture helps the reader's understanding. We added this:

Fig. 2. Hooked -end steel fibres used.

  1. How much % by vol. fraction was the steel fibres used?

It was used a 0.44%vol. of steel fibres. The following sentence has been completed “The concrete mix proportions were obtained by the Fuller method, adding a 0.44% by volume of steel fibres… “

  1. Eq. 1, CMOD, what does CMOD term stand for? in the text above, there is crack opening (COD) but no mention of CMOD.

Indeed, not only had we forgotten to explain the term correctly, at some point the terms had got mixed up. In general, COD refers to the device used to measure the crack opening, while CMOD (Crack Mouth Opening displacement) is the measured parameter. The difference between the COD and the CMOD was explained in lines 141 to 143 “To measure the CMOD a COD extensometer was used. Two metal pieces were fixed to the two sides of the crack, in the centre of it, and an extensometer was fitted, taking the specimens to breakage and recording the Load-COD data.” Also, the whole document has been checked for those cases where by mistake these two terms have been mixed up.

  1. For equations, please provide what which term means, like in Eq. 2, σ = 3 * F * L / 2 * b * h^2, please write, "where F = ...., L = .... etc" for more clarity.

Thank you very much for the comment, a paragraph has been added indicating what is each variable: “σ=stress; F=load applied; L= distance between the two lower cylinders; b= specimen width; h= distance from the base to the beginning of the notch.”

  1. I am not sure, if its only which me or not, in Figures 13, 14 and 16, the legend in yellow colour is hard to read, if possible could the authors can the colour from yellow to another.

Thank you very much for the observation, indeed the yellow lines were not easy to interpret. So the yellow lines were changed to green.

  1. Please conduct a comprehensive English proofread to remove the grammatical mistakes.

A comprehensive English proofread to remove the grammatical mistakes by a native has been performed.

All in all, this article is worth publishing in Materials journal as it fits the scope of the journal and also contributes to the scientific advances.
